# Therapeutic Effects of Lactoferrin in Ocular Diseases: From Dry Eye Disease to Infections

**DOI:** 10.3390/ijms21186668

**Published:** 2020-09-12

**Authors:** Aldo Vagge, Carlotta Senni, Federico Bernabei, Marco Pellegrini, Vincenzo Scorcia, Carlo E Traverso, Giuseppe Giannaccare

**Affiliations:** 1University Eye Clinic of Genoa, DiNOGMI—University of Genoa IRCCS Ospedale, Policlinico San Martino, Viale Benedetto XV, 5, 16132 Genova (GE), Italy; mc8620@mclink.it; 2Ophthalmology Unit, S.Orsola-Malpighi University Hospital, University of Bologna, 40138 Bologna, Italy; c.senni3@gmail.com (C.S.); federico.bernabei89@gmail.com (F.B.); marco.pellegrini@hotmail.it (M.P.); 3Department of Ophthalmology, University Magna Græcia of Catanzaro, 88100 Catanzaro, Italy; vscorcia@libero.it (V.S.); giuseppe.giannaccare@gmail.com (G.G.)

**Keywords:** lactoferrin, antimicrobial peptides, ocular surface, dry eye, biofilm, viral infections, Sars-CoV-2

## Abstract

Lactoferrin is a naturally occurring iron-binding glycoprotein, produced and secreted by mucosal epithelial cells and neutrophils in various mammalian species, including humans. It is typically found in fluids like saliva, milk and tears, where it reaches the maximum concentration. Thanks to its unique anti-inflammatory, antioxidant and antimicrobial activities, topical application of lactoferrin plays a crucial role in the maintenance of a healthy ocular surface system. The present review aims to provide a comprehensive evaluation of the clinical applications of lactoferrin in ocular diseases. Besides the well-known antibacterial effect, novel interest has been rising towards its potential application in the field of dry eye and viral infections. A growing body of evidence supports the antimicrobial efficacy of lactoferrin, which is not limited to its iron-chelating properties but also depends on its capability to directly interact with pathogen particles while playing immunomodulatory effects. Nowadays, lactoferrin antiviral activity is of special interest, since lactoferrin-based eye drops could be adopted to treat/prevent the new severe acute respiratory syndrome coronavirus type 2 (SARS-CoV-2) infection, which has conjunctivitis among its possible clinical manifestations. In the future, further data from randomized controlled studies are desirable to confirm the efficacy of lactoferrin in the wide range of ocular conditions where it can be used.

## 1. Introduction

The ocular surface system is an essential component of vision. It includes the cornea, conjunctiva, tear film, eyelids and lacrimal and meibomian glands, all components linked functionally by the continuity of the epithelia, innervation and endocrine, vascular and immune systems [1]. As a part of a complex morphofunctional unit, each constituent of this system acts to provide, protect and maintain a smooth refractive surface on the cornea, whose avascularity and transparency enable light to proceed through the lens onto the retina. As a direct interface between the eye and the external environment, the ocular surface is constantly exposed to potentially harmful agents, like microbes and toxic substances, which place it at risk of destructive immunological reactions. Hence, loss of the outer eye’s structural integrity may follow, leading to various degrees of visual impairment.

Host defenses at the level of the ocular surface include a combination of mechanical, anatomical and immunological mechanisms. Both conjunctival and corneal cells provide an epithelial barrier to prevent pathogen entrance, and concomitantly secrete cytokines to activate immune defenses against microbial invasion. Besides mediating reflexive blinking movements, corneal nerves play a key role in the maintenance of corneal trophism, and thanks to neuropeptide release they have the capability to induce cytokine activity and the subsequent neutrophil inflow [2,3,4]. Cooperation between innate and acquired arms of the immune system is crucial to keep the homeostatic balance of the ocular surface united once pathogen assault occurs, since the structural integrity of the whole system is required to ensure its normal functioning. Among the effectors of innate immunity, complement comprises a series of bioactive compounds such as enzymes, opsonins, anaphylatoxins and chemotoxins that are activated by means of a chain reaction and contribute to the onset of corneal inflammation. In addition, neutrophils and macrophages synergistically act to protect ocular surface epithelia from pathogen invasion. The former play a key role in phagocytosis and microbial killing, while the latter feature both phagocytic and antigen-presenting capabilities and adjuvate inflammatory reactions by secreting cytokines [5,6]. On the other hand, interferons (IFNs) and natural killer (NK) cells activate each other in response to viral infection. In particular, IFNs stimulate the production of major histocompatibility complex (MHC) class I molecules and proteins, which allow virally infected cells to be recognized by T cells, whereas IFN-mediated activation of NK lymphocytes leads to the targeting and subsequent elimination of virus-hosting cells [7]. Not least, tears also act as a source of unspecific antimicrobial substances like lysozyme, lactoferrin, lipocalins and secretory phospholipase A2, which are effective in counteracting the invasion and colonization of microorganisms at the ocular surface [8].

Although several antimicrobial drugs may be used to treat infections, concerns are rising in regard to the increasing prevalence of drug-resistant microbes (in particular for antibiotics) and the limited efficacy of the available compounds. Thus, it is important to look for and to develop additional drugs that could either overcome the issue of drug resistance, or to act synergistically when combined with other treatments. Antimicrobial peptides that are constitutively represented in human fluids and tissues may provide the basis for the development of new therapeutic agents to be successfully applied in the management of ocular surface infections.

The purpose of the present review is to sum up available evidence regarding the use of lactoferrin, a natural peptide commonly found in tears, in the setting of ocular surface diseases, with special attention being paid to its role as a new potential antiviral agent.

## 2. Lactoferrin

Formerly known as lactotransferrin, lactoferrin is an iron-binding glycoprotein, a member of the transferrin protein family alongside serum transferrin, ovotransferrin, melanotransferrin and the inhibitor ocarbonic anhydrase [9]. First isolated in bovine milk, it is naturally produced and secreted by mucosal epithelial cells and neutrophils in various mammalian species, including humans. Lactoferrin is particularly found in fluids like saliva, milk and tears, and expressed in various organs including the mammary gland, uterus, kidney and brain [10,11,12,13]. It consists of 80 kDa of the glycosylated protein of about 700 amino acids with high homology among species, and features some important biological properties such as antioxidant, anticancer, anti-inflammatory and antimicrobial activities (Figure 1). The single polypeptide chain is assembled into two symmetrical globular N- and C-terminal lobes, each one containing two domains, referred to as N1 and N2, or C1 and C2, which enclose a deep cleft where the iron-binding site is located [14].

Tear lactoferrin represents 25% of total tear proteins, with an average concentration in healthy subjects of 1.42 mg/mL. It is mostly secreted by the main lacrimal gland, with both epithelial cells and meibomian acini contributing to its final tear levels [15]. Aging as well as the presence of a dysfunctional ocular surface, as occurs in the setting of keratitis and conjunctivitis of different aetiologies, result in a decrease of tear lactoferrin, thus exposing patients affected by these conditions to a higher risk of infection.

## 3. Lactoferrin and Dry Eye

Topical application of lactoferrin has been shown to reduce irradiation-induced corneal epithelial damage in mice models, as well as to promote corneal wound healing after alkali-burn injury [16,17]. Furthermore, previous studies have reported a significant correlation between low levels of tear lactoferrin and the development of both dry eye disease (DED) and chronic meibomitis [18,19,20] (Figure 2A). These two common disorders of the ocular surface system share a degree of mutual pathophysiology, since markers of inflammation and oxidative stress are present in both conditions [21]. Indeed, a decreased volume and an altered qualitative composition of tear film, together with excessive tear evaporation, lead to the creation of a hyperosmolar environment, which then initiates both inflammatory and oxidative cascades, resulting in impaired epithelial proliferation and differentiation [22]. The rationale of the use of lactoferrin in the setting of DED derives from its capacity to directly address the vicious cycle of the disease, especially the underlying inflammation and oxidative stress. In particular, thanks to its iron-chelating ability, lactoferrin provides oxygen free radical and hydroxyl scavenging activities, thus inhibiting pro-inflammatory and tissue damaging effects of reactive oxygen species (ROS). On the other hand, lactoferrin attenuates excessive inflammation in host responses to pathogens by inhibiting classical complement activation and by downregulating inflammatory mediators such as tumor necrosis factor (TNF)-alpha, interleukins (ILs)-1, -6 and -8, the intercellular adhesion molecule (ICAM)-1 and CD14 [23] (Figure 1). In a study by Dogru et al., patients supplemented with oral lactoferrin showed ameliorated dry eye symptoms and tear film stability [24]; another study reported its efficacy in improving ocular surface parameters, such as tear break-up time and the Schirmer test, in patients affected by dry eye induced by cataract surgery [25]. Furthermore, locally applied lactoferrin was able to restore corneal epithelial integrity in a rabbit model of dry eye, suggesting the potential use of lactoferrin eye drops for treating DED [26].

## 4. Lactoferrin and Infections

The activity of lactoferrin against microbial invasion was the first to be discovered, and to date is also the most widely studied. On one hand, it sequesters free iron, thus exerting a bacteriostatic effect by removing a key substrate for bacterial growth; it also damages bacteria by binding iron ions, leading to oxidative stress, impaired membrane permeability and cell lysis [27,28]. On the other hand, through an iron-independent mechanism, it exerts a bactericide effect by directly interacting with the bacterial wall surface [29]. In particular, thanks to its unique macroscopic array, consisting of large cationic surface patches, lactoferrin easily binds to the anionic Lipid A, which is the innermost region of the three regions of the lipopolysaccharide (LPS) of Gram-negative bacteria. By means of such interaction, lactoferrin is able to directly damage the bacterial membrane and prevent neutrophil inflow, subsequently leading to an inhibition of superoxide anion production and to the downregulation of inflammatory response. [30,31]. Starting from this evidence, lactoferrin has been shown to efficiently prevent the pathological ingrowth of several bacteria, including *Escherichia coli*, *Haemophilus influenzae*, *Bacillus subtilis*, *Streptococcus* spp., *Staphylococcus* spp. and *Pseudomonas* spp. [32,33,34,35,36]. No less importantly, it has been reported that lactoferrin could act against *Pseudomonas aeruginosa* by inhibiting both colonization and biofilm formation on contact lens surfaces, and by increasing susceptibility to topically applied antibiotics [37,38]. A similar effect was also noted in the setting of fungal keratitis, where lactoferrin was able to prevent biofilm formation over contact lenses.

As a multifunctional protein, lactoferrin also exhibits efficacy in the setting of viral infectious processes against both naked and enveloped human and animal pathogenic viruses. Indeed, comparative studies between formula-fed and breast-fed children showed that the latter experienced fewer infections with rotavirus, respiratory syncytial virus (RSV) or vesicular stomatitis virus [39]. Further studies have highlighted the activity of lactoferrin against other common viral particles, including cytomegalovirus (CMV), herpes simplex virus (HSV), human immunodeficiency virus (HIV), hepatitis C virus (HCV), poliovirus (PV), parainfluenza virus (PIV), human papillomavirus (HPV) and adenovirus [40]. In particular, the antiviral activity of lactoferrin lies in the early phase of infection, when it prevents virus entry into host cells. Such inhibition of viral infection mediated by lactoferrin occurs either via binding to heparan sulphate glycosaminoglycan cell receptors or by directly interacting with viral particles. The former mechanism explains the great efficacy of lactoferrin against HSV infection and HSV cell-to-cell spread, while the latter appears to be crucial in the attempt to prevent HCV and HIV entry into host cells. Additionally, in vitro evidence shows that lactoferrin is able to act against viral infection in a dose-dependent manner. By binding to sulphate glycosaminoglycan cell receptors, lactoferrin prevents the viral concentration on the cell surface and the subsequent specific interaction between the spike viral protein and the angiotensin-converting enzyme 2 (ACE2) receptor, which is able to hook the viral terminals and facilitate the entry into the cell [41]. In addition, the antiviral action of lactoferrin may be partly related to its immunomodulatory effects, consisting in interferon (IFN)-alpha upregulation, enhanced activity of NK lymphocytes, promotion of T cell precursors’ maturation into competent helper cells, induced differentiation of immature B cells into efficient antigen presenting cells and stimulation of antibody response [42,43].

To date, poor evidence is available about the application of lactoferrin in the setting of viral infections of the ocular surface. Special attention has been paid to HSV keratitis, which can lead to corneal blindness in the most severe cases (Figure 2B). HSV has been reported to be present in the trigeminal ganglion of nearly 100% of patients over the age of 60 at autopsy. It might affect every layer of the cornea, and characteristically presents with dendritic lesions at the fluorescein staining test. Once the virus is reactivated from latency within the sensory ganglion, it potentially results in corneal scarring, necrosis and decreased sensation with the onset of neurotrophic keratopathy. This condition is a degenerative disease affecting corneal nerves that amplifies the detrimental effect of HSV infection on the ocular surface by reducing both protective reflexes and the release of trophic neuromodulators that are essential for the vitality, metabolism and wound healing of surface tissues. As a result, impaired corneal innervation leads to spontaneous epithelial breakdown, persistent epithelial defects and ulceration with subsequent onset of vision issues [44]. Both in vivo and ex vivo anti-herpetic activities of lactoferrin have been investigated. The binding of lactoferrin to virus receptors on target cells prevented virus attachment and entry, whereas orally administered lactoferrin increased the production of Th1 cytokines, including IFN-gamma, IL-12 and IL-18, which may adjuvate host protection against HSV infection [45,46,47,48,49,50].

Siciliano et al. demonstrated that anti-HSV activity is specifically mediated by the N-lobe, and also that synthetic lactoferrin-derived peptides could exert antiviral activity, although they exhibited less efficacy when compared with the native protein [49]. Based on immunofluorescence microscopy studies, an additional effect of lactoferrin on viral infection was noted. Indeed, the few viral particles that were internalized appeared to have a delayed intracellular trafficking [51]. This finding suggests that lactoferrin exerts its beneficial effects at different stages of the HSV cycle. Moreover, the transport of lactoferrin-filled endosomes along microtubules not only delays intracellular HSV trafficking by competing with the microtubule-mediated transport of the virus, but could also prevent viral replication, which requires an intact microtubule network to succeed [51]. In a study by Fujihara et al., the topical administration of 1% lactoferrin, prior to virus inoculation, suppressed HSV-1 infection in the mouse cornea [52]. Furthermore, the key role of such a bioactive compound in the management of viral colonization has been confirmed by another study, where a certain lactoferrin polymorphism was shown to be associated with the susceptibility of developing HSV keratitis [53].

By adopting similar mechanisms, lactoferrin may address adenovirus, which is the main causative factor of epidemic keratoconjunctivitis (Figure 2C).

This is a highly contagious condition associated with marked inflammation, conjunctival hyperemia and edema, and symptoms of irritation, tearing, blurry vision and light sensitivity. Its inappropriate diagnosis and treatment may lead to the development of pseudomembranous conjunctivitis, with resultant scarring and symblepharon formation, which alter the local homeostatic balance contributing to the onset of a vicious circle of ocular surface disease.

Taken together, these findings support the key role of lactoferrin in ensuring the homeostatic balance of the ocular surface, and highlight the multiple activities of this molecule, ranging from the recovery of epithelial integrity to antibacterial and antiviral effects. Nowadays, this last activity appears to be of special interest, since the current coronavirus disease 2019 (COVID-19) pandemic has conjunctivitis among its possible clinical manifestations. Indeed, it has been reported that lactoferrin takes part in the host immune response against severe acute respiratory syndrome coronavirus (SARS-CoV) invasion by enhancing NK cell activity, stimulating neutrophil aggregation and adhesion and preventing the interaction between the viral particle and its cell receptors, represented by heparan sulfate proteoglycans (HSPGs) [54]. Starting from this evidence, similar mechanisms have been proposed for SARS-CoV-2 (COVID-19) infection, for which lactoferrin has additionally shown the capability of inhibiting in vitro viral replication [55].

## 5. Conclusions

Nature provides us with several examples of the use of micronutrients with a medical purpose. As physiological constituents of human tissues, substances deriving from either food intake or nutraceutical products can act as bioactive compounds and influence both the morphology and function of the ocular surface system components by taking part in several metabolic cellular pathways aiming at preserving a homeostatic balance. In this regard, topical lactoferrin answers the need to develop new treatments to target infections at the ocular surface. This glycoprotein has some important advantages. First of all, it is easily found in nature and exhibits multiple beneficial effects on ocular tissues. Its efficacy as an antimicrobial agent has been widely reported, and interest regarding the application of lactoferrin in the setting of viral infections is growing. In this field, lactoferrin could be adopted either as a single therapeutic agent or in combination with other treatments to maximize efficacy. Further data from randomized controlled studies are desirable to confirm the efficacy of lactoferrin in the wide range of ocular conditions where it can be used.

## Figures and Tables

**Figure 1 ijms-21-06668-f001:**
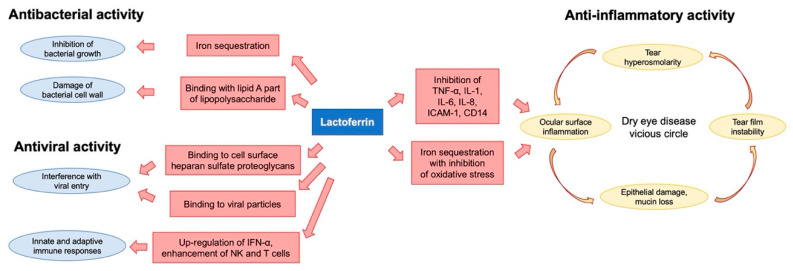
Flowchart showing the mechanisms of action of lactoferrin in ocular diseases. Iron sequestration is the basis of the bacteriostatic effect of lactoferrin. The molecule also has a direct bactericidal activity thanks to its binding with lipid A domains of the bacterial lipopolysaccharide. The binding to viral surface components as well as to heparansulfate proteoglycans inhibits the virus–host cell interaction and is responsible for the antiviral activity. Moreover, lactoferrin regulates innate and adaptive immune responses against infections. The effects of lactoferrin on the vicious circle of dry eye disease are related to the downregulation of numerous pro-inflammatory cytokines as well as to the anti-oxidative activity.

**Figure 2 ijms-21-06668-f002:**
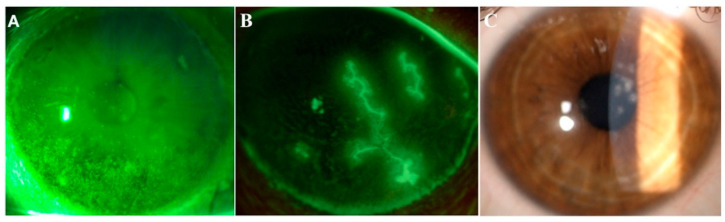
Images from three representative patients affected by dry eye, herpetic keratitis and adenoviral keratoconjunctivitis. Representative images of ocular conditions that may benefit from the use of lactoferrin: dry eye (**A**), herpetic keratitis (**B**) and epidemic keratoconjunctivitis (**C**). (a) Slit lamp photograph of the cornea of a patient with dry eye after the instillation of 20 μL of unpreserved 2% sodium fluorescein and use of the yellow filter to enhance the staining details. The epithelial damage is visible with fluorescein staining as multiple punctate epithelial erosions scattered over the corneal surface, in particular in the lower sectors. (b) Slit lamp photograph of the cornea of a patient with herpetic keratitis after the instillation of 20 μL of unpreserved 2% sodium fluorescein and use of the yellow filter to enhance the staining details. Classical dendritic epithelial defects are visible with positive fluorescein staining. (c) Slit lamp photograph of a patient with adenoviral conjunctivitis showing multifocal sub-epithelial (stromal) corneal infiltrates.

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
