# Peer review of "Therapeutic Effects of Lactoferrin in Ocular Diseases: From Dry Eye Disease to Infections"

_ijms, 2020, doi:10.3390/ijms21186668_

Round 1

Reviewer 1 Report

Despite references explain how lactoferrin is important for dry eye and other ocular diseases, I'd like to see more evidences about the mechanism of action. A figure (with legend) might be added in order to better understand the drug's effect.

Reviewer 2 Report

The author comprehensively review the functions and potential effects of lactoferrin on the ocular surface, including maintenance of ocular surface homeostasis, and antimicrobial efficacy, especially in the setting of bacterial and viral infections. This article is well written, clear and reasonable, and will not cause misunderstanding. Therefore, in addition to correcting some minor typing errors, there is only one recommendation for the author:

  1. Recommendation: please summarize the important lactoferrin mechanisms and effects on dry eye and different infections (especially for ocular infections) in a flowchart. which also can be used in the graphic abstract.
  2. Some typing errors:
    • line 99 on page 5: mammalians species => mammalian species
    • line 112 on page 6: patients affected by these condition at a higher risk of infection => patients affected by these conditions at a higher risk of infection
    • line 154 on page 7: Escherichia coli, Haemophilus influenzae, Bacillus subtilis, Streptococcus spp, => Escherichia coli, Haemophilus influenzae, Bacillus subtilis, Streptococcus spp.,
    • line 155 on page 8: Staphylococcus spp and Pseudomonas spp => Staphylococcus spp. and Pseudomonas spp.
    • line 156 on page 8: that lactoferrin could act against Pseudomonas Aeruginosa => that lactoferrin could act against Pseudomonas aeruginosa
